# VLMaterial: Procedural Material Generation with Large Vision-Language Models

**Beichen Li**[1], **Rundi Wu**[2], **Armando Solar-Lezama**[1], **Changxi Zheng**[2], **Liang Shi**[1], **Bernd Bickel**[3,4], **Wojciech Matusik**[1]

[1]MIT CSAIL, [2]Columbia University, [3]ETH Zürich, [4]Google Research

## Abstract

Procedural materials, represented as functional node graphs, are ubiquitous in computer graphics for photorealistic material appearance design. They allow users to perform intuitive and precise editing to achieve desired visual appearances. However, creating a procedural material given an input image requires professional knowledge and significant effort. In this work, we leverage the ability to convert procedural materials into standard Python programs and fine-tune a large pre-trained vision-language model (VLM) to generate such programs from input images. To enable effective fine-tuning, we also contribute an open-source procedural material dataset and propose to perform program-level augmentation by prompting another pre-trained large language model (LLM). Through extensive evaluation, we show that our method outperforms previous methods on both synthetic and real-world examples.

## 1 Introduction

High-quality materials are essential to creating digital 3D assets in computer graphics applications such as video games, movies and AR/VR. Image-based materials use several texture maps to represent material attributes (*e.g.,* albedo, normal, and roughness). While extensively used in the rendering pipeline, they are not easily editable and are limited by the image resolution. As an alternative, procedural materials have become particularly popular in modern 3D content creation workflows due to an interpretable, controllable, and resolution-independent material representation. In modern 3D design and content creation tools, *e.g.,* Blender (Blender, 2024a) and Adobe Substance 3D (Adobe, 2024), a procedural material is typically defined as a directed computation graph of pre-defined functions. More specifically, the material graph consists of nodes representing texture generators or filtering operators and edges representing the information flow, producing the required material attributes of an appearance shading model (see Figure 1 right for an example). Users can easily edit the material by adjusting the parameters of each node or manipulating the edges.

Complex procedural material graphs can contain dozens of nodes, which requires professional skills and significant effort to create manually. This has motivated the research on inverse procedural material modeling, *i.e.,* automatically generating a procedural material to match the appearance of a captured or rendered input image. Early works retrieve an artist-created material graph from an online repository and focus on optimizing its node parameters (Hu et al., 2019; Shi et al., 2020; Hu et al., 2022a; Li et al., 2023; Hu et al., 2022b). MatFormer (Guerrero et al., 2022) first proposed to generate the entire material graph by training a multi-stage transformer-based model that sequentially generates nodes, edges, and parameters. Hu et al. (2023) later extended the model for conditional generation from text or image prompts. Instead of training a similar, custom network from scratch, we explore the fact that procedural material graphs in Blender (Blender, 2024a) can be transpiled into standard Python programs and fine-tune a pre-trained large vision-language model (VLM) to generate such programs directly.

A straightforward approach is to directly prompt pre-trained VLMs to generate procedural material programs from images. However, we find that existing commercial VLMs (*e.g.,* GPT-4o-mini (OpenAI, 2024b)) struggle on this task (see Table 1) as procedural material programs are domain-specific and underrepresented in the training data of VLMs. Therefore, we fine-tune a pre-trained VLM

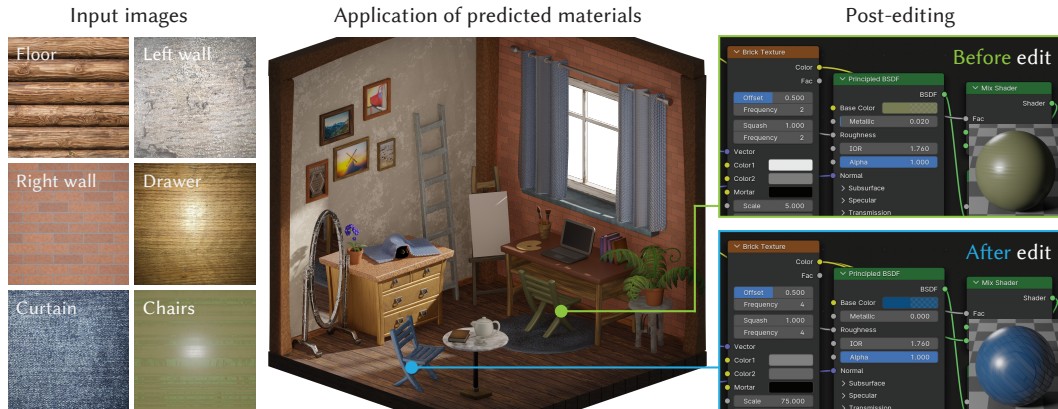

Figure 1: Given single input images (left), our model generates corresponding procedural materials (top right) that can be directly applied to a 3D scene (middle). The generated materials can be easily edited, *e.g.,* by changing some node parameters, to achieve desired visual appearance (bottom right).

on pairs of images and procedural material programs. The scarcity of publicly available procedural materials, however, poses a fundamental challenge. Inspired by FunSearch (Romera-Paredes et al., 2024), we leverage pre-trained VLMs to perform program-level data augmentation followed by carefully designed node parameter augmentation. This data augmentation strategy extends our dataset from $\sim 1.6$K examples to $\sim 550$K examples, significantly improving the matching quality of predicted materials.

Through extensive evaluation, we show that our method outperforms previous single-image procedural material generation methods on both synthetic and real-world examples. As shown in Figure 1, our generated procedural materials can be directly applied to a 3D scene with full editability. Furthermore, previous works are developed for proprietary software, *e.g.,* Adobe Substance 3D (Adobe, 2024), thus precluded from releasing training data. We contribute the first open-source procedural material dataset in Blender (Blender, 2024a) to promote future research in this area. Our dataset and code are available at `https://github.com/mit-gfx/VLMaterial`.

## 2    RELATED WORK

**Inverse procedural material modeling.**    Most previous work on material generation focus on synthesizing image-based texture maps (Li et al., 2017; 2018; Deschaintre et al., 2018; Gao et al., 2019; Guo et al., 2020; 2021; Henzler et al., 2021; Vecchio et al., 2021; 2024a;b; Zhou & Kalantari, 2021; Martin et al., 2022; Zhou et al., 2022; 2023). Inverse procedural material modeling, instead, aims to generate procedural material programs from captured images. Early works along this direction assume a known procedural material graph and use either learning-based or optimization-based approaches to find the best matching parameters (Hu et al., 2019; Shi et al., 2020; Hu et al., 2022a; Li et al., 2023; Hu et al., 2022b). For example, Hu et al. (2019) trains a CNN for each material graph to regress the best parameters that match the input image. MATch (Shi et al., 2020) uses gradient descent to optimize the parameters by making the graph execution in Substance (Adobe, 2024) differentiable, which is later extended to cover most available material node types for end-to-end differentiability (Hu et al., 2022a; Li et al., 2023). More recent works learn to predict or refine both the graph structure and parameters of procedural materials (Guerrero et al., 2022; Hu et al., 2023; Huang et al., 2024). MatFormer (Guerrero et al., 2022) flattens the material graphs into custom token sequences and trains three Transformers (Vaswani et al., 2017) to generate nodes, edges, and parameters for each node sequentially. Hu et al. (2023) subsequently conditions MatFormer on input text descriptions or images. Li et al. (2024a) further applies reinforcement learning (RL) fine-tuning to improve parameter predictions. Instead of training a custom network from scratch, we leverage that procedural material graphs in Blender (Blender, 2024a) can be transpiled into Python programs and fine-tune a VLM for image-conditioned program generation.

**Visual program synthesis.**    Our work is also related to visual program synthesis, *i.e.,* inferring the underlying programs from input visual signals. Prior works under this topic span many domains, such as synthesizing drawing programs from images (Johnson et al., 2017; Ellis et al., 2018; 2019; Tian et al., 2019; Ellis et al., 2023), converting raster images into SVGs (Reddy et al., 2021; Carlier et al., 2020; Rodriguez et al., 2023), reconstructing CSG trees from 3D shapes (Sharma et al., 2018; Du et al., 2018; Kania et al., 2020; Jones et al., 2020; Uy et al., 2022), and inferring CAD construction programs from 3D shapes (Wu et al., 2021; Xu et al., 2021; Li et al., 2022; Xu et al., 2023). In contrast to some of these works that define their own domain specific language, we focus on generating procedural material programs that are already widely used in the 3D design industry.

**LLMs for procedural modeling.**    Recent Large Language Models (LLMs) like GPT-4o (OpenAI, 2024a), LLaMA (Touvron et al., 2023) and Gemini (Team et al., 2023) have demonstrated impressive results in code generation and program synthesis (Romera-Paredes et al., 2024; Olausson et al., 2023). As a procedural modeling system essentially defines a domain specific language, some recent works explore procedural modeling using LLMs (Sun et al., 2023; Huang et al., 2024; Yang et al., 2024; Yamada et al., 2024). 3D-GPT (Sun et al., 2023) uses an LLM as a planner to combine pre-defined procedural generators to create a 3D scene corresponding to the given text description. BlenderAlchemy (Huang et al., 2024) leverages VLMs to iteratively refine a procedural modeling program to achieve the design intent of the user. These works all use the zero-shot reasoning ability of LLMs to modify existing procedural modeling programs. Instead, we focus on generating procedural material programs from scratch based on the input images and find that state-of-the-art VLMs fail to perform this task reliably via zero-shot prompting. As a solution, we fine-tune a VLM on our collected procedural material dataset with a carefully designed data augmentation scheme and show a significant performance improvement.

## 3    BACKGROUND

Procedural materials are functional programs that produce photorealistic material appearances on objects after rendering, typically visualized as *node graphs* in industrial rendering software. A procedural material $M$ is mathematically equivalent to a directed acyclic computation graph $G = (\mathcal{V}, \mathcal{E})$. Each node $v \in \mathcal{V}$ represents a parameterized texture generation or filtering operator $x_{\text{out}}^+ = f(x_{\text{in}}^*, \phi)$ that receives zero or multiple input arguments $x_{\text{in}}^*$ and generates one or more outputs $x_{\text{out}}^+$. Its functionality is additionally controlled by a set of heterogeneous node parameters $\phi = (p_1, p_2, \cdots)$ that are either continuous (*e.g.,* floats, vectors, colors, and color ramps) or discrete (*e.g.,* integers, categories, and Booleans) by nature. Each edge $e \in \mathcal{E}$ represents the data flow from an output connector of one node to an input connector of another node. Notably, adjacent nodes with several input and output connectors can be linked by multiple edges. A user can easily edit a procedural material in a graphical interface by adding or removing nodes, changing edge connections, and tweaking node parameters, to produce a range of appearances.

Among several industry-standard procedural material systems in modern 3D software (Adobe, 2024; Blender, 2024a; Epic, 2024), we tackle the inverse design problem of Blender procedural materials due to its well-established open-source community and mature Python API. Blender procedural materials utilize texture nodes to generate primitive noise or geometric patterns (*e.g.,* bricks, checker, Voronoi, and fractal noise), shader nodes to specify surface properties (*e.g.,* roughness, normals, and metallicity), and converter nodes for color mapping and general mathematical operations. The node graph of an example Blender procedural material predicted by our model is shown in Figure 2. Furthermore, Blender allows users to package a node subgraph into a custom macro-node with user-specified inputs and outputs, referred to as a *node group*. Utilizing various node groups, procedural materials in Blender can represent sophisticated appearances with compact graph structures.

## 4    METHOD

Figure 2 illustrates the workflow of our method. Given an input image capturing the material appearance on a flat surface, we use a fine-tuned large VLM to predict a Blender procedural material in Python code. Executing the code in Blender yields the shader node graph of the material, which is rendered into the output image under a point light source. The procedural materials predicted by our model preserve full editability albeit under the constraint of pre-defined node and parameter types from Blender (Blender, 2024b) and users. Following previous works (Shi et al., 2020; Hu et al.,

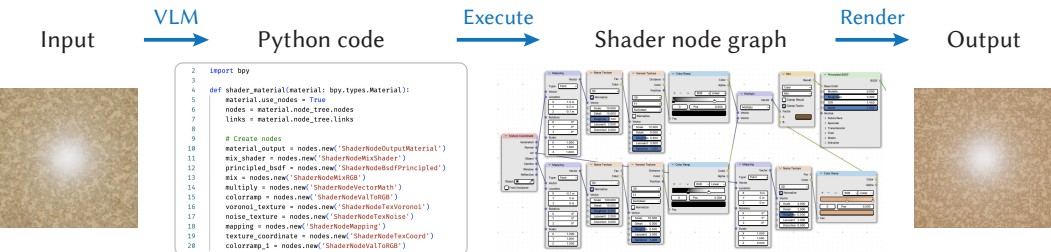

Figure 2: Workflow overview. Given an input image, we use our fine-tuned VLM to predict a procedural material in the form of Python code. Executing the code in Blender yields a user-editable shader node graph, whose rendered appearance closely matches the input image.

2023; Li et al., 2023), we aim to achieve a close perceptual match with the input image instead of a pixel-perfect reproduction.

Fine-tuning a large VLM requires a comprehensive procedural material dataset with diverse material appearances and graph structures. Prior work (Hu et al., 2023) has developed such a dataset for Adobe Substance 3D (Adobe, 2024) materials but their dataset can not be released due to proprietary intellectual property. To address the lack of an open-source counterpart, we specially curate a dataset containing over 550K Blender procedural materials (Section 4.1 and 4.2). The remaining subsections describe our fine-tuning process (Section 4.3) and post-optimization algorithm (Section 4.4).

## 4.1 DATA COLLECTION

We first collected 3,663 free Blender materials from three online sources: 1) 2,411 materials in BlenderKit[1], an online repository of Blender 3D assets; 2) the 60 base materials of Infinigen (Raistrick et al., 2023), a procedural 3D scene generation framework; 3) 1,192 materials from individually published Blender procedural material packs[2]. These materials are manually created by experienced technical artists and cover a wide variety of material categories. Then, we conduct the following curation process before adding each material to training data:

**Node graph clean-up.** We run a depth-first search (DFS) from the material output node to identify nodes that participate in output appearance calculation, removing other redundant nodes and edges. The material is dropped if a valid output node does not exist or the node graph is too large (with more than 30 nodes). In addition, each node group in the material is treated as a custom node type distinguished by its functional signature. To prevent the VLM from struggling to memorize too many custom node types, we iteratively expand the smallest node groups in these materials into corresponding subgraphs until reaching the graph size limit.

**Program validation.** We subsequently transpile the remaining materials into Python code and render the resulting appearances. This step filters out materials with excessively long code sequences (more than 2,048 tokens after tokenization) or degenerate renderings, including empty images, uniform colors, or textures lacking meaningful spatial features. We heuristically measure the complexity of a material using the compressed JPEG file size of its rendered image. By sorting the materials with descending JPEG file sizes, we observe that most materials below 12KB exhibit undesirable texture quality. We thus apply this heuristic threshold to discard unwanted, meaningless materials from the dataset.

## 4.2 DATA AUGMENTATION

The scarcity of high-quality artist-created procedural materials has been a fundamental challenge for training generative models (Hu et al., 2023). Blender materials are no exception—the curation process above leaves 1,640 materials in our dataset, which we found severely inadequate for VLM fine-tuning. Hu et al. (2023) proposes to create appearance variations of existing materials by randomly perturbing node parameters. While this effectively introduces color and pattern changes,

---

[1] https://www.blenderkit.com
[2] https://heypictures.gumroad.com/l/zwhai;https://maroc777772.gumroad.com/l/oaehcd

---

**Algorithm 1** MCMC-based local parameter search.

---

1: $i \leftarrow 0$, $M \leftarrow \tilde{M}$, $l \leftarrow L(I, \tilde{I})$
2: $M^* \leftarrow M$, $l^* \leftarrow l$                 $\triangleright$ Initialize the result
3: **while** $i < N_{\text{iters}}$ **do**
4:   Randomly sample $M$'s node parameters to obtain $M'$    $\triangleright$ See Appendix A.3 for details
5:   $l' \leftarrow L(I, R(M'))$
6:   **if** $l' < l$ **then**
7:     $M \leftarrow M'$, $l \leftarrow l'$              $\triangleright$ Accept a better solution
8:   **else**
9:     Set $M \leftarrow M'$, $l \leftarrow l'$ at a small probability $p_{\text{acc}}$   $\triangleright$ Otherwise, accept stochastically
10:   **end if**
11:   **if** $l < l^*$ **then**
12:     $M^* \leftarrow M$, $l^* \leftarrow l$             $\triangleright$ Update the best solution
13:   **end if**
14:   $i \leftarrow i + 1$
15: **end while**

---

the sampled variants retain the same node graph structure, which hardly prevents the model from overfitting to a limited set of graph structures.

Our data augmentation strategy combines the augmentation of graph structures and parameters. On the graph structure level, we augment the 1.6K artist-created materials with diverse program structures using an approach similar to the evolutionary workflow in FunSearch (Romera-Paredes et al., 2024). Specifically, we manually select a subset of 870 complex materials with rich texture detail from the artist-created materials into a sample pool. The evolution process involves selecting random program pairs from the sample pool and performing "genetic crossovers" using a commercial pre-trained LLM (GPT-4o-mini), which is prompted to generate novel programs using elements from the selected programs. The LLM can restructure the example programs and modify their node parameters as needed. We then validate the LLM-generated programs and add executable programs to the sample pool, where each program pair contributes up to four valid programs out of 20 trials. Our evolutionary workflow produces 50.4K programs with distinct structures, more than $30\times$ the number of artist-created materials. We provide the complete LLM prompt in Appendix A.1. After graph structure augmentation, we conduct parameter-space augmentation following Hu et al. (2023), creating another $10\times$ more program variations from random parameter perturbations (see Appendix A.1 for implementation details). The augmented dataset eventually comprises over 550K materials, more than $330\times$ the initial size.

### 4.3 VLM FINE-TUNING

Our augmented procedural material dataset enables the adaptation of pre-trained VLMs to image-conditioned procedural material generation tasks. We employ the latest version of LLaVA-NeXT (Li et al., 2024b) from the LLaVA model family (Liu et al., 2023b;a; 2024), a series of open-source multi-modal LLMs trained on public-domain visual question answering (VQA) data. The model architecture consists of a CLIP ViT/L-14 vision encoder (Radford et al., 2021), a LLaMA 3 8B language decoder (Dubey et al., 2024), and an MLP projector that bridges the embedding spaces of the encoder and decoder networks. Additional training-related details are available in Appendix A.2.

### 4.4 POST-OPTIMIZATION

After predicting a plausible procedural material with the VLM, we can further improve the perceptual match against the input image by refining its node parameters. This post-optimization step is commonly used in previous works concerning Adobe Substance 3D materials (Shi et al., 2020; Hu et al., 2022a; 2023; Li et al., 2023). In particular, they leverage a differentiable implementation of the node graph to backpropagate analytical gradients from a perceptual loss based on the Gram matrices of VGG features (Gatys et al., 2015). However, Blender procedural materials currently do not have differentiable counterparts, nor is it feasible to train individual differentiable proxies (Hu et al., 2022a) for hundreds of built-in and custom node types.

Table 1: A quantitative comparison of our method with several baselines on in-distribution (Blender) and out-of-distribution (Substance and real images) datasets. For a fair comparison, the numbers for our approach here are before post optimization. Red: best score, Orange: second best score.

| Dataset | Method | Style loss ↓ | SWD ↓ | CLIP ↑ | Program correctness ↑ |
|---------|--------|--------------|-------|--------|----------------------|
| Blender | GPT-4o-mini | 0.041 | 3.478 | 0.728 | 0.294 |
| | Cond. MatFormer | 0.034 | 2.828 | 0.686 | — |
| | BlenderAlchemy | 0.027 | 2.381 | 0.777 | — |
| | Nearest Neighbor | 0.026 | 2.123 | 0.778 | — |
| | Ours | 0.019 | 1.760 | 0.856 | 0.911 |
| Substance | GPT-4o-mini | 0.039 | 3.722 | 0.680 | 0.227 |
| | Cond. MatFormer | 0.033 | 2.366 | 0.736 | — |
| | BlenderAlchemy | 0.029 | 2.727 | 0.690 | — |
| | Nearest Neighbor | 0.027 | 2.546 | 0.720 | — |
| | Ours | 0.026 | 2.283 | 0.762 | 0.890 |
| Real images | GPT-4o-mini | 0.035 | 4.252 | 0.642 | 0.248 |
| | Cond. MatFormer | 0.028 | 2.872 | 0.684 | — |
| | BlenderAlchemy | 0.025 | 2.682 | 0.700 | — |
| | Nearest Neighbor | 0.021 | 2.487 | 0.700 | — |
| | Ours | 0.025 | 2.417 | 0.722 | 0.870 |

Instead, we opt for a gradient-free, stochastic local parameter search based on the Markov Chain Monte Carlo (MCMC) algorithm (Geyer, 1992; Diaconis, 2009), which has been applied to inductive program synthesis (Schkufza et al., 2013). Let $\tilde{M}(\Phi)$ be a predicted procedural material with node parameters $\Phi$ that yields an output rendering image $\tilde{I} = R(\tilde{M})$, $L(I, \tilde{I})$ be the perceptual loss function between the input image $I$ and the prediction $\tilde{I}$. Our MCMC algorithm translates into the pseudocode in Algorithm 1. We run MCMC sampling for each predicted material for $N_{\text{iters}} = 200$ iterations, accepting a worse sample in each iteration at a small probability $p_{\text{acc}} = 0.05$.

## 5 EXPERIMENT

### 5.1 EVALUATION SETUP

**Datasets.** We evaluate our method on in-distribution and out-of-distribution test images collected from three different sources: 1) 44 synthetic materials from *Blender* (Blender, 2024a), randomly selected and separated from our training dataset (before augmentation); 2) 64 synthetic materials from *Substance* (Adobe, 2024), which are randomly sampled from the evaluation set of (Hu et al., 2023); 3) 64 real photographs gathered from Shi et al. (2020) and Zhou et al. (2023), captured using smart phone cameras. We note that the node graph system of Adobe Substance 3D is more expressive than Blender and capable of creating more intricate material appearances. Therefore, although the *Substance* test set comprises synthetic images, we consider them an out-of-distribution test set harder than *Blender*.

**Baselines.** We compare our method to four baselines: GPT-4o-mini (OpenAI, 2024b), nearest neighbor retrieval, Conditional MatFormer (Hu et al., 2023) and BlenderAlchemy (Huang et al., 2024). For GPT-4o-mini (OpenAI, 2024b), we directly query the model with the input image and a carefully designed prompt (see Appendix A.4). For nearest neighbor retrieval, we retrieve the nearest material from our original training dataset based on the style loss (Gatys, 2015) between its rendered image and the input image. Conditional MatFormer (Hu et al., 2023) trains a custom transformer on 446K procedural material programs (after parameter augmentation) from Substance Source (Adobe, 2024). Since its code is not publicly available, we asked the authors to run their trained model on our test sets. BlenderAlchemy (Huang et al., 2024) starts with an initial procedural material program and iteratively queries a VLM to modify the program to better match the input image. We run their method starting from the nearest neighbor retrieval result and process VLM queries with GPT-4o-mini. We compare all methods without performing post-optimization.

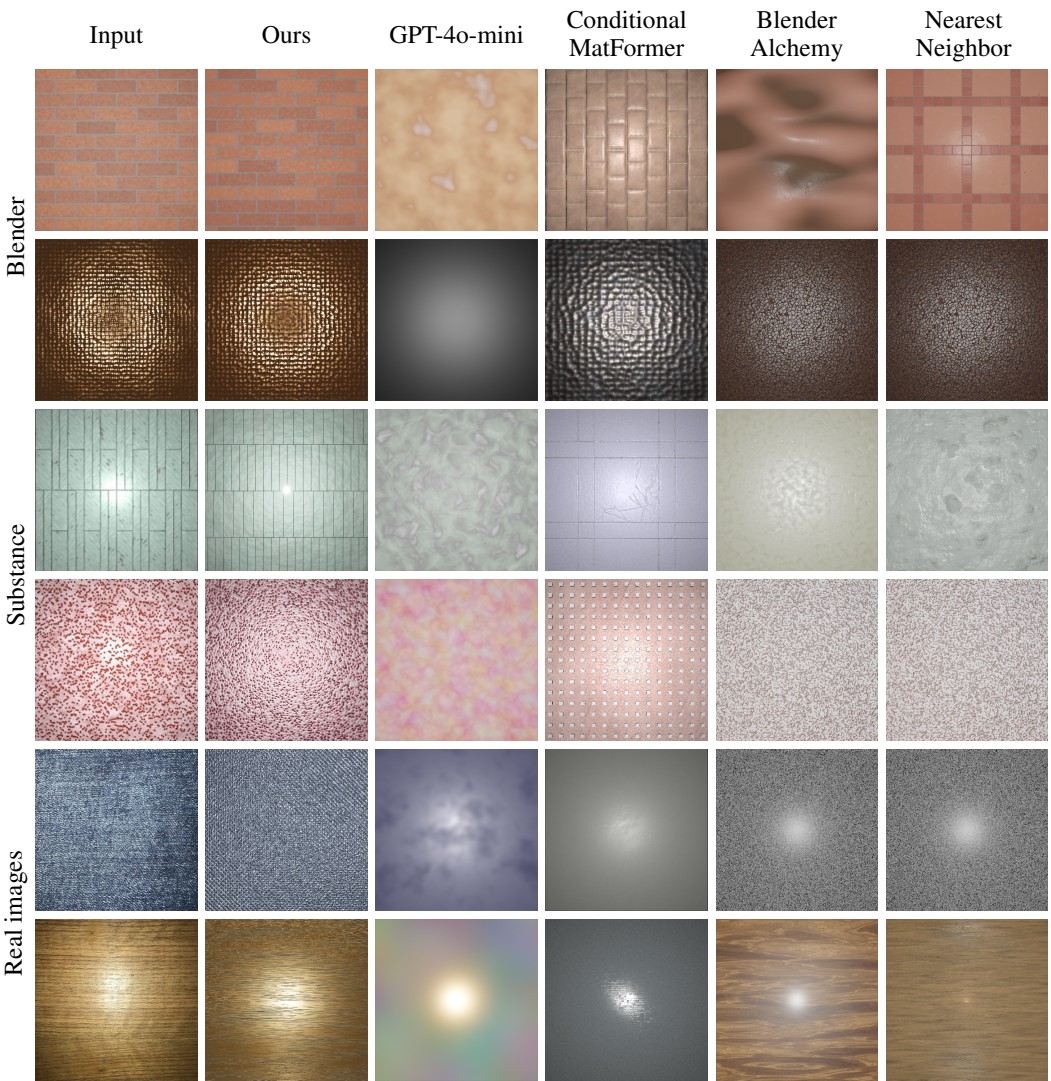

Figure 3: A qualitative comparison of our method with several baselines. The input images are shown in the leftmost column. Re-rendered images from our generated procedural material programs best match the input.

**Evaluation protocol.** Similar to Hu et al. (2023), we generate at most $N$ programs using our model and stop once obtaining $K < N$ valid programs. We then pick the program with the lowest style loss (Gatys, 2015) between the input and rendered images as the final result. Following Section 4.1, a generated program is considered valid if it successfully executes and produces an output rendering above the 12KB JPEG file size threshold. For a fair comparison, we evaluate GPT-4o-mini and Conditional MatFormer using the same protocol. We use $N = 50$ and $K = 20$ for the following quantitative and qualitative results. Additional comparison results under a tighter sample budget ($N = 20$, $K = 4$) are included in Appendix B.

**Evaluation metric.** To measure the perceptual similarity between the rendered image of a generated program and the input image, we use three metrics—style loss (Gatys, 2015), sliced Wasserstein distance (SWD) (Bonneel et al., 2015), and CLIP cosine similarity (Radford et al., 2021). The style loss is a widely adopted metric for texture distance. Following prior work (Hu et al., 2023), we compute the style loss using the L1 difference of Gram matrices of VGG features (Gatys et al., 2015) plus the L1 difference of $16 \times 16$ downsampled images (weighted by 0.1). SWD calculates the distance between two images based on the optimal transport theory (Villani et al., 2009), which

Table 2: Ablation study. - *parameter aug.* removes the graph parameter augmentation, and - *structure aug.* further removes the graph structure augmentation. + *post opt.* optimizes graph parameters in our generated programs using MCMC.

| Dataset | Method | Style loss ↓ | SWD ↓ | CLIP ↑ | Program correctness ↑ |
|---|---|---|---|---|---|
| Blender | - structure aug. | 0.032 | 2.757 | 0.750 | 0.603 |
| | - parameter aug. | 0.022 | 1.965 | 0.839 | 0.897 |
| | Ours | 0.019 | 1.760 | 0.856 | 0.911 |
| | + post opt. | 0.015 | 1.701 | 0.851 | — |
| Substance | - structure aug. | 0.033 | 3.084 | 0.727 | 0.487 |
| | - parameter aug. | 0.030 | 2.482 | 0.744 | 0.767 |
| | Ours | 0.026 | 2.283 | 0.762 | 0.890 |
| | + post opt. | 0.020 | 2.184 | 0.748 | — |
| Real images | - structure aug. | 0.030 | 3.308 | 0.688 | 0.342 |
| | - parameter aug. | 0.028 | 2.682 | 0.707 | 0.713 |
| | Ours | 0.025 | 2.417 | 0.722 | 0.870 |
| | + post opt. | 0.018 | 2.309 | 0.732 | — |

has been demonstrated for neural texture synthesis (Heitz et al., 2021). In addition, the CLIP metric measures the semantic difference of textures as a complement.

Aside from appearance matching, we also evaluate the model's ability to produce valid procedural material programs. Inheriting the evaluation protocol above, assuming we run the model $n \leq N$ times and get $k \leq K$ valid programs for a test image, we define a *correctness* score as $k/n$, i.e., the fraction of valid program predictions. This metric only applies to our method and GPT-4o-mini, since Conditional MatFormer represents procedural materials as customized token sequences and implements an inference-time grammar checker to ensure the generated program is executable.

## 5.2 RESULTS

We report quantitative comparison results in Table 1, and show qualitative examples in Figure 3 and Figure 5. GPT-4o-mini (OpenAI, 2024b) struggles to produce valid material programs. Conditional MatFormer (Hu et al., 2023) performs reasonably well on the Substance dataset (on which it was trained) but shows limited generalization to the other test sets. Nearest neighbor retrieval obtains competitive quantitative scores, which agrees with the findings in previous work (Hu et al., 2023). However, most retrieved materials are visually sub-optimal against the input. Blender-Alchemy (Huang et al., 2024) often produces results that do not deviate much from the starting programs. Moreover, it incurs a considerable running cost since each program generation requires dozens of multi-modal queries to the GPT model.

Our method outperforms the baselines on all three test datasets. The generated appearances are structurally and semantically close to the input images. To demonstrate the practical usability of our generated materials, we apply them to 3D meshes and show the rendering results in Figure 1. We also conduct a user study among professional artists and domain researchers to validate how the generated materials substitute an artist's creation process (see Appendix C for details).

## 5.3 ABLATION STUDY

In Table 2 and Figure 4, we compare our method against ablated versions without data augmentation and validate the post-optimization algorithm. Both graph structure augmentation and parameter augmentation are crucial to improving the matching quality on in-distribution and out-of-distribution test sets. Specifically, structure augmentation guides the model to predict more accurate texture patterns, while parameter augmentation enhances the matching of color and other appearance details. As for program correctness, our model has a significantly higher probability of generating valid programs after fine-tuning using the fully augmented dataset. In addition, our results are further improved by the post-optimization step using MCMC local parameter search. For generated pro-

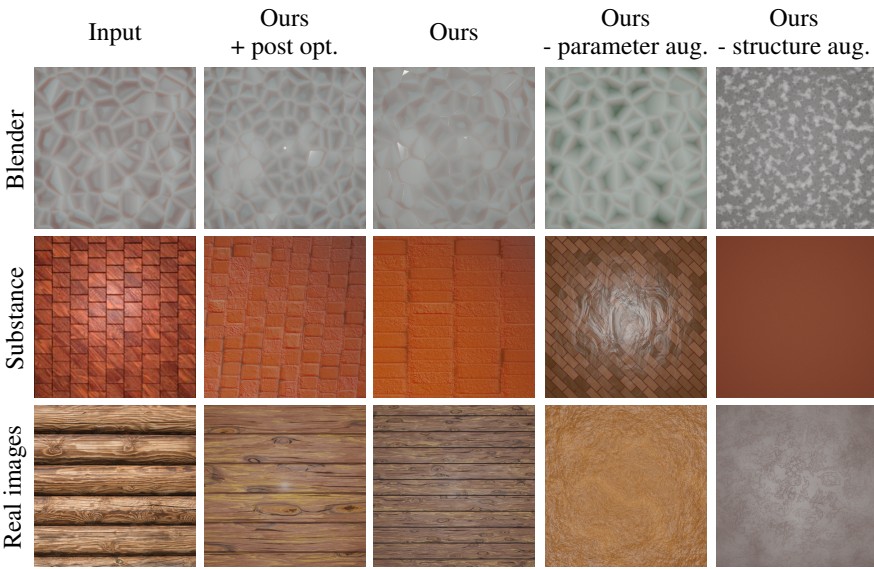

Figure 4: Visual examples for ablation study.

grams with sufficient capacity to represent the input image, the post-optimization algorithm refines the overall color and spatial structures, achieving noticeably better alignment with the input.

## 6  DISCUSSION AND CONCLUSION

Figure 7 demonstrates a few failure cases of our method where the intricate texture pattern of the input image exceeds the expressiveness of existing Blender node types (including artist-created node groups) within a reasonable node graph size (*e.g.,* we use a limit of 30 nodes). In such cases, our model may struggle to predict the best material graph structure to approximate the input texture patterns. Extending the node graph system with custom node groups from a broader range of artist-created materials will alleviate the limitation in expressiveness. An alternative is to increase the graph size limit and the VLM's context window, although this requires additional engineering efforts in data augmentation to generate more feature-rich materials using an LLM.

Future works may address the challenge of generating complex visual programs using VLMs from several orthogonal directions. Including more artist-created materials and performing data augmentation using next-generation LLMs with advanced chain-of-thought (CoT) reasoning may improve the complexity and diversity of training data. Further, it is possible to represent Blender procedural materials with a compact domain-specific language (DSL) based on handcrafted rules or library learning. This will produce considerably shorter transpiled programs than Python and simplify VLM fine-tuning. Moreover, an inference-time syntax checker can be implemented for the DSL to mask invalid tokens and guarantee the correctness of generated programs. Finally, incorporating visual feedback through RL fine-tuning (Li et al., 2024a) is an effective approach to enhance visual quality. Employing an image-space reward function bridges the gap between token accuracy and visual similarity, potentially resulting in better generalization to unseen images.

In conclusion, we present a single-image procedural material generation method that fine-tunes a pre-trained large VLM to predict Blender procedural materials in Python code format. Our approach harnesses the embedded knowledge of the VLM through domain specialization, enabled by the first open-source procedural material dataset with more than 550K image-program pairs. We report superior perceptual matching quality on out-of-distribution synthetic and real input images, outperforming existing models trained on proprietary materials or leveraging zero-shot VLM inference. We hope the release of our dataset and model will benefit future research on visual program synthesis and inverse procedural graphics.

ACKNOWLEDGEMENTS

This work was partially funded by an unrestricted gift from Google.

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

## A   IMPLEMENTATION DETAILS

### A.1   DATA AUGMENTATION

**Graph structure augmentation.**   We prompt GPT-4o-mini to generate novel programs using the template below based on pairs of example programs.

```
You are familiar with creating procedural materials using
Blender's Python API. You will be given a Python code block
delimited by triple backticks, which contain two functions
that define the node trees of two procedural materials in
Blender. The functions are named 'shader_material_1' and
'shader_material_2', respectively. Your task is to write
a Python function 'shader_material' that creates a new
material based on the provided functions. Write your code
following the guidelines below.

Code template:
```python
import bpy

def shader_material(material: bpy.types.Material):
    material.use_nodes = True
    nodes = material.node_tree.nodes
    links = material.node_tree.links

    # Create nodes
    # YOUR CODE HERE

    # Create links to connect nodes
    # YOUR CODE HERE

    # Set parameters for each node
    # YOUR CODE HERE
```

Rules:
1. Create no more than 30 nodes. Only use node types and
parameter fields referenced in the provided functions. Change
parameter values as needed.
2. Make sure your code can be correctly executed in
Blender 3.3. Refer to the Blender Python API documentation
for valid node types and parameters.
3. Try to generate materials with complex and semantically
meaningful appearances. Avoid generating materials that lack
structure or look too similar to the provided examples.
4. Follow the format and coding style of the example
functions.
Do not add new code blocks or comments.
5. Simply reply with code. Exclude any additional text or
explanations.
```

**Parameter augmentation.**   Given a procedural material with a prescribed graph structure, we randomly sample its node parameters around their initial values to generate appearance variations. The collection of sampled parameters includes 1) unoccupied input and output connectors of each node with well-defined default values, *e.g.,* the color input to a principled BSDF shader node; 2) node-specific attributes, such as the operation type of a math node and the interpolation option of a color ramp node. Although our sampling strategy is similar to Hu et al. (2023), we develop a native im-

plementation for Blender's procedural material system, which applies different sampling algorithms contingent on whether a node parameter is inherently continuous or discrete.

- For a continuous parameter, *i.e.,* a float, vector, color, or color ramp, we refer to its definition in the associated node and its value statistics across the procedural material dataset to determine a feasible range. The parameter is then uniformly sampled within the intersection of the feasible range and a $\pm 25\%$ interval around the initial value. Vector or array elements are sampled independently from one another. Considering that the relative interval may be insufficient for small initial values, we enforce $\pm 0.05$ as an absolute, minimal sampling interval. Specifically, color parameters are sampled in the hue-saturation-value (HSV) color space instead of the conventional RGB space. We uniformly sample hue channels from the entire $[0, 1]$ range and sample the other channels within a $\pm 0.5$ interval. This encourages diverse color tones in sampled materials.

- We divide discrete parameters into two separate scenarios. An integer parameter is treated as continuous and rounded to the nearest integer after uniform sampling. Categorical and Boolean parameters, in contrast, do not have the notion of absolute values or intervals. We thus uniformly sample each parameter from its value range at a probability of 0.25.

An important caveat of sampling all node parameters at once is the additional program length. In Blender Python programs, explicit assignments are unnecessary for node parameters remaining at default values. Sampling all node parameters away from their default values will accumulate dozens of lines of code, creating excessively long code sequences. We prevent this issue by sampling node parameters holding default values with a probability of 0.2. If the resulting program exceeds the token length limit (*i.e.,* 2,048 LLaMA 3 tokens), we randomly reset some parameters back to default values to reduce the code length. The resulting program must pass the validation (Section 4.1) before being incorporated into training data.

## A.2 VLM FINE-TUNING

For supervised fine-tuning, we pair the input image with the following user prompt to the VLM and tokenize the ground-truth Python program using LLaMA 3's tokenizer.

```
<image>
Write a Python function with Blender API to create a material
node graph for this image.
```

The trainable modules include the MLP projector and an array of LoRA adapters (Hu et al., 2021) applied to all attention-related linear layers in the LLaMA 3 model (with $r = 8$, $\alpha = 32$, and a 0.05 dropout probability), amounting to 40M trainable parameters in total.

We use an AdamW optimizer (Loshchilov, 2017) with a 1e-4 learning rate and a cosine annealing schedule. The initial linear warmup period accounts for 3% of training steps. To effectively utilize GPU memory, we apply FlashAttention-2 (Dao, 2023) and train the model in BF16 precision on $8\times$ NVIDIA H100 80GB GPUs using DeepSpeed ZeRO-3 (Rasley et al., 2020; Rajbhandari et al., 2020). Each GPU accommodates a batch size of 4 training samples, resulting in an overall batch size of 32. The fine-tuning process lasts 5 epochs over three days.

## A.3 POST-OPTIMIZATION

We largely follow the same node parameter sampling method as data augmentation (Appendix A.1) for MCMC sampling. However, we only randomly sample 10% of available node parameters in each iteration. Furthermore, we constrain the relative uniform sampling interval for ranged parameters to $\pm 20\%$ (including colors) and the sampling probability of categorical parameters to 0.2. These practices prompt the MCMC algorithm to make smaller steps in each iteration, improving the likelihood of converging to a better solution. We also ignore the program length limit during post-optimization. Therefore, the node parameters are no longer differentiated based on default values.

A.4 EVALUATION

We apply the same user prompt as training to our fine-tuned VLM for inference. We use the following prompt to evaluate the GPT-4o-mini baseline.

```
You are familiar with creating procedural materials using
Blender's Python API. You will be given an image that
describes a material appearance. Your task is to write a
Python function 'shader_material' that creates a Blender
procedural material to match the appearance of the image
when rendered on a flat surface. Write your code following
the guidelines below.

Code template:
```python
import bpy

def shader_material(material: bpy.types.Material):
    material.use_nodes = True
    nodes = material.node_tree.nodes
    links = material.node_tree.links

    # Create nodes
    # YOUR CODE HERE

    # Create links to connect nodes
    # YOUR CODE HERE

    # Set parameters for each node
    # YOUR CODE HERE
```

Rules:
1. Create no more than 30 nodes.
2. Make sure your code can be correctly executed in Blender 3.3.
Refer to the Blender Python API documentation for valid node
types and parameters.
3. Simply reply with code. Exclude any additional text or
explanations.
```

# B ADDITIONAL RESULTS AND DISCUSSIONS

**Qualitative comparisons.** We provide additional qualitative examples and ablation study results in Figure 5 and Figure 6, respectively. Figure 7 shows some failure cases, a detailed discussion around which is included in Section 6.

**Quantative comparison in a limited budget.** We also compare our method and baselines quantitatively using a more limited sample budget during inference in Table 3, where the prediction result is selected from up to $K = 4$ valid samples out of $N = 20$ trials. Our method is affected by the variance in token predictions, which encourages exploration during inference. Fine-tuning the VLM with RL using an image similarity reward will likely reduce the entropy of token predictions and improve generalization in budget-limited scenarios. Nevertheless, RL fine-tuning should carefully balance exploration and exploitation to prevent convergence to local minima.

**Inference time consumption.** We report the average inference time of our method and other baseline methods in Table 4. Specifically, for our method and Conditional MatFormer, the time consumption is measured on an NVIDIA H100 80GB GPU at the best throughput. The remaining baselines

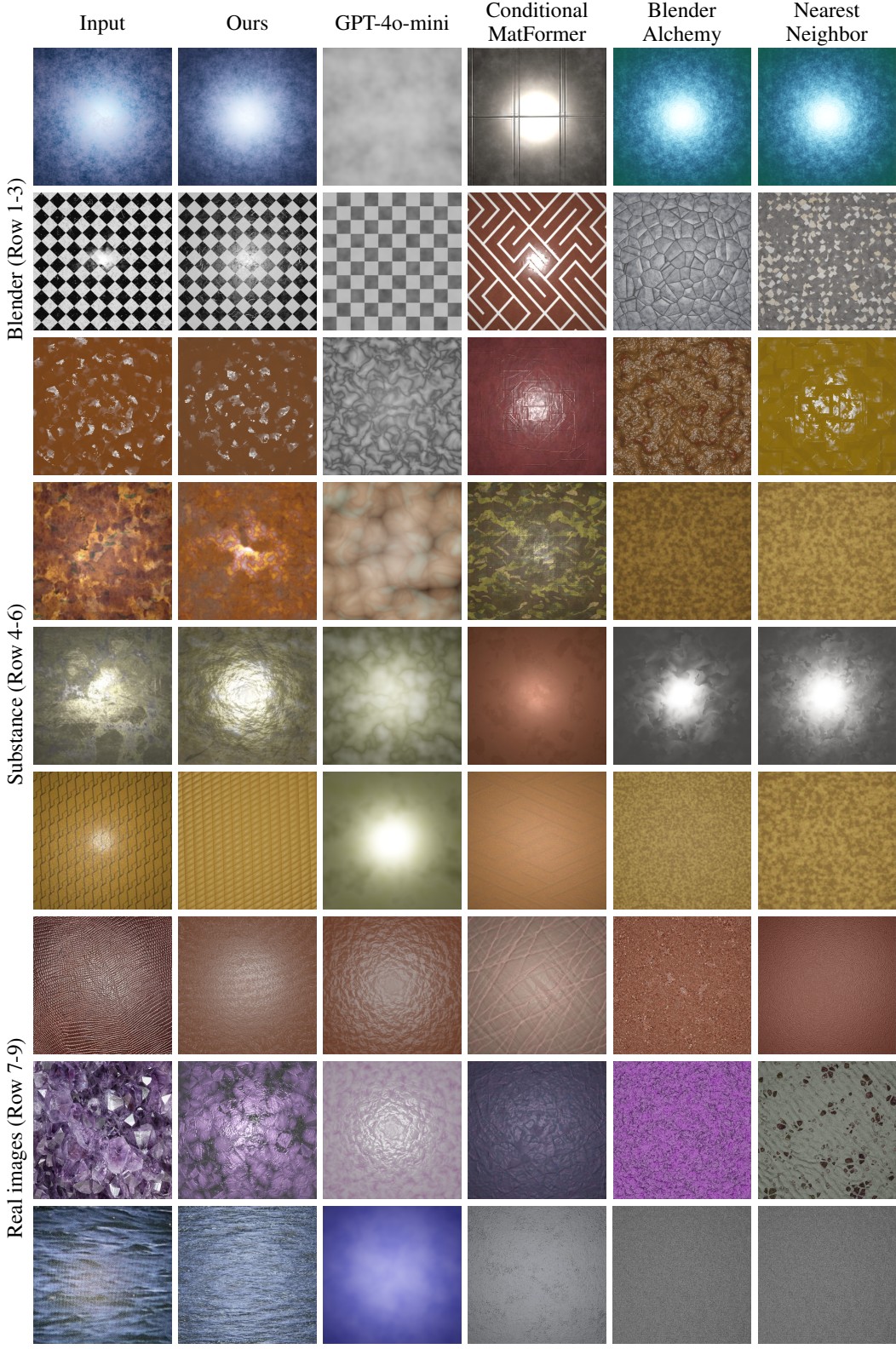

Figure 5: More qualitative comparisons with baselines.

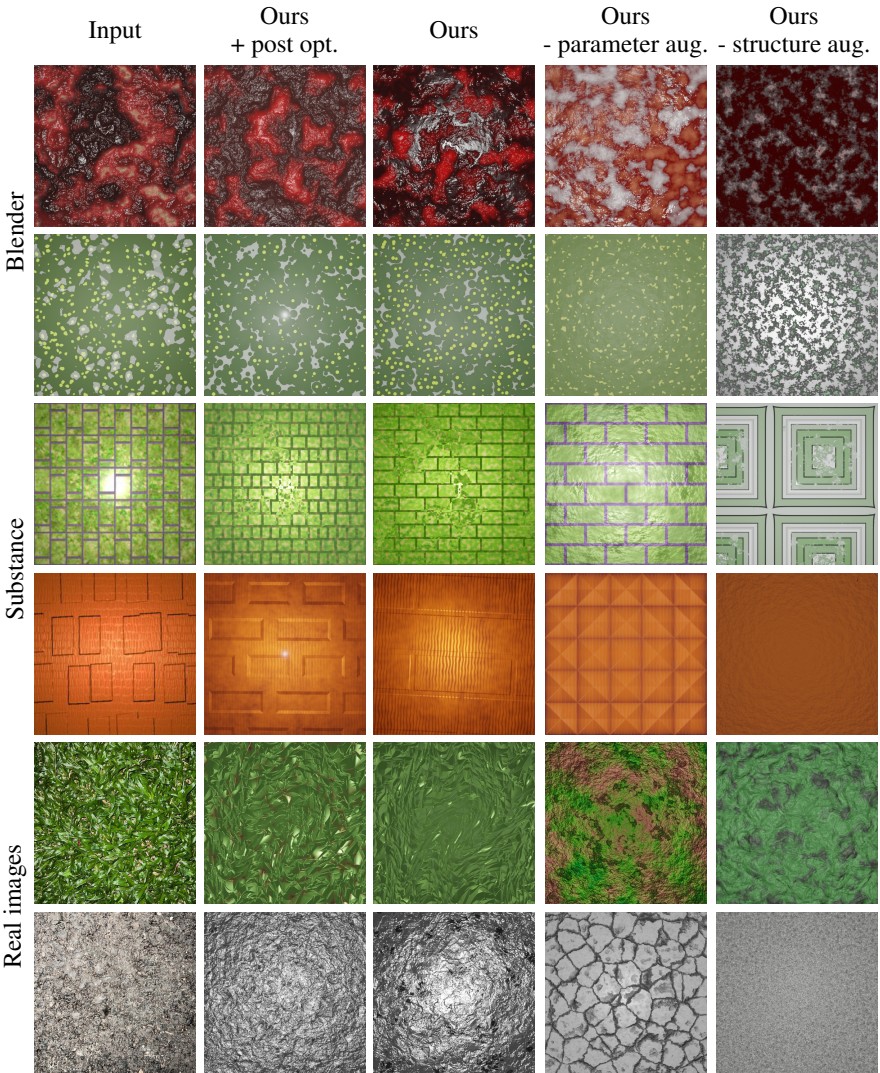

Figure 6: More examples for ablation study.

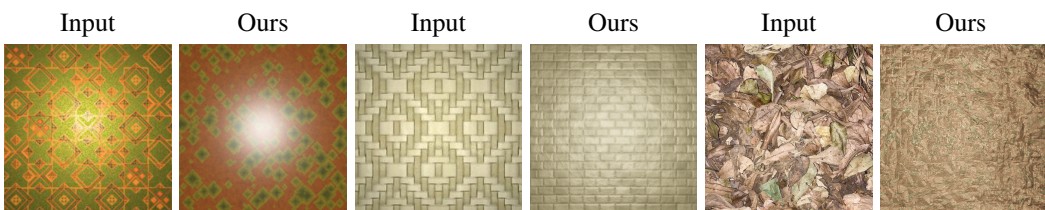

Figure 7: Failure cases. The first two examples (Column 1-4) come from the *Substance* test set. The last example (Column 5-6) is from the real image test set.

are timed on an AMD Ryzen 7 5800X 8-core CPU with parallel requests to the OpenAI API. Conditional MatFormer has a high inference throughput due to a small network size. However, even after selecting the top-5 graphs from 150 random samples, their predictions still require a costly post-optimization to match the input (Hu et al., 2023).

**Performance impact of material complexity.** We plot the average style loss and program correctness for VLM-generated materials with varied complexity in Figure 8. The materials are divided

Table 3: Comparison between our method and baselines under a more constrained sample budget ($N = 20, K = 4$). The results of BlenderAlchemy and Nearest Neighbor are copied from Table 1.

| Dataset | Method | Style loss ↓ | SWD ↓ | CLIP ↑ | Program correctness ↑ |
|---|---|---|---|---|---|
| Blender | GPT-4o-mini | 0.044 | 3.618 | 0.716 | 0.343 |
| | Cond. MatFormer | 0.042 | 3.019 | 0.682 | — |
| | BlenderAlchemy | 0.027 | 2.381 | 0.777 | — |
| | Nearest Neighbor | 0.026 | 2.123 | 0.778 | — |
| | Ours | 0.025 | 1.978 | 0.833 | 0.926 |
| Substance | GPT-4o-mini | 0.042 | 3.799 | 0.678 | 0.234 |
| | Cond. MatFormer | 0.040 | 2.281 | 0.742 | — |
| | BlenderAlchemy | 0.029 | 2.727 | 0.690 | — |
| | Nearest Neighbor | 0.027 | 2.546 | 0.720 | — |
| | Ours | 0.031 | 2.431 | 0.750 | 0.926 |
| Real images | GPT-4o-mini | 0.040 | 4.369 | 0.636 | 0.241 |
| | Cond. MatFormer | 0.037 | 2.869 | 0.690 | — |
| | BlenderAlchemy | 0.025 | 2.682 | 0.700 | — |
| | Nearest Neighbor | 0.021 | 2.487 | 0.700 | — |
| | Ours | 0.030 | 2.566 | 0.716 | 0.880 |

Table 4: The inference time consumption of our method and other baselines.

| Method | Avg. time (s) | #Samples | Avg. time per sample (s) |
|---|---|---|---|
| GPT-4o-mini | 7.5 | 20 | 0.38 |
| Cond. MatFormer | 4.1 | 20 | 0.21 |
| BlenderAlchemy | 252.4 | 32 | 7.89 |
| Ours | 47.9 | 20 | 2.40 |

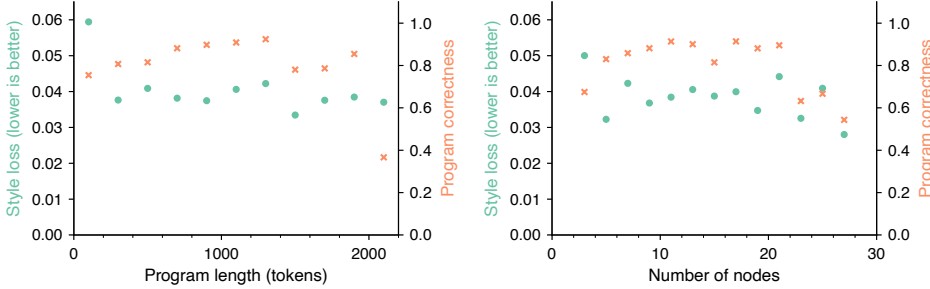

Figure 8: The average style loss and program correctness of VLM-generated materials with varied material complexity. The complexity is measured using program lengths in tokens and node graph sizes, respectively.

into separate bins based on the transpiled program length and the number of nodes. Most materials with moderate complexity show comparable style losses and correctness. Interestingly, both metrics are worse for simple materials with fewer than 4 nodes, largely due to invalid or suboptimal choices of custom node groups. Program correctness also declines for materials with more than 22 nodes or 2,000 tokens, implying a higher chance of errors in more intricate materials. As mentioned in Section 6, implementing inference-time syntax checking to mask out invalid tokens can guarantee the correctness of generated programs.

Table 5: Guidelines for interpreting the usability ratings of algorithm-generated procedural materials in our user study. Intermediate ratings indicate preferences between two adjacent descriptions.

| Rating | Description |
|---|---|
| 1 | Barely useful. It is better to create the material manually. |
| 4 | Partially reusable as a template. Many sections need to be reworked. |
| 7 | A good starting point, requiring careful but not too cumbersome post-editing. |
| 10 | An ideal solution. Very few edits are necessary. |

Table 6: User study on the practical usability of procedural materials generated with our fine-tuned VLM and BlenderAlchemy (Huang et al., 2024). We report the average 2AFC preference score and usability rating over different test sets. Numbers after the "±" symbol are standard deviations.

| Preference (Ours over BlenderAlchemy) | Blender | Substance | Real images | **Average** |
|---|---|---|---|---|
| Visual quality | 92% | 95% | 86% | 91% |
| Node graph quality | 92% | 91% | 98% | 94% |
| **Usability Rating** (1-10) | | | | |
| BlenderAlchemy | $4.1 \pm 1.7$ | $3.9 \pm 1.9$ | $4.5 \pm 1.5$ | $4.2 \pm 1.7$ |
| Ours | $7.9 \pm 1.2$ | $5.6 \pm 1.7$ | $6.9 \pm 1.0$ | $6.8 \pm 1.3$ |

## C  USER STUDY

To investigate the practical value of VLM-generated procedural materials to an artist's creation process, we hosted a user study among professional artists who specialize in Blender material graphs and researchers with sufficient domain expertise. The participants are tasked with comparing the materials generated from our method and BlenderAlchemy (Huang et al., 2024) for 12 randomly selected test images (4 images from each test set). Each comparison involves the following items:

- **Preference in visual quality.** The participant chooses which method ("A" or "B") produces a better visual match with the input image. We adopt the two-alternative forced choice (2AFC) approach and randomly shuffle the results to conceal the source algorithms.

- **Preference in node graph quality.** In the same 2AFC setting, we show the participant a snapshot of the generated material node graph from each method and ask them to determine which graph requires simpler user post-editing to match the input image.

- **Usability rating.** The participant rates the usability of each generated material on a scale from 1 to 10 based on how much it substitutes an artist's creation process. The interpretations of usability ratings are provided in Table 5.

We collected 16 responses from 8 professional artists and 8 researchers, respectively. Then, we calculated the average preference percentage for our method over BlenderAlchemy and the average usability rating for both methods in Table 6. In more than 90% of the test cases, participants prefer our generated materials over BlenderAlchemy in visual and node graph quality. Our method achieves an average usability rating of 6.8, indicating that artists generally consider VLM-generated materials efficient starting points for post-editing. In addition, some artists praised our method for the interesting results and noted how our method alleviates the core issues with complex shader graphs compared with image-based textures. They also appreciated the better reusability of simpler graph structures in production environments.

