# OpenReview forum: "VLMaterial: Procedural Material Generation with Large Vision-Language Models"
_ICLR.cc/2025/Conference — ICLR 2025 Spotlight_

### Official Review · Reviewer_bStb · 2024-10-31

**Soundness:** 3
**Presentation:** 3
**Contribution:** 3
**Rating:** 6
**Confidence:** 4

**Summary:**

The paper presents a novel approach to procedural material generation from input images using a large, fine-tuned vision-language model (VLM). This method transforms material generation into a text-to-program problem, converting material graphs to Python code for Blender's API. The paper introduces an open-source procedural material dataset and proposes a two-tier data augmentation strategy to enhance training, achieving a substantial improvement in program accuracy and visual quality over existing methods.

Several contributions:
1. VLM Fine-Tuning for Procedural Materials: The authors fine-tune a VLM to generate procedural material node graphs in Python code format based on input images, addressing the limitation that existing VLMs lack training data specific to procedural materials.

2. Open-Source Dataset for Procedural Materials: The authors compile an open-source dataset of 550,000 procedural material samples for training and evaluation, combining real artist-created Blender materials with augmented data.

3. Post-Optimization Algorithm: A gradient-free, Markov Chain Monte Carlo (MCMC) approach refines the generated material node graphs to better match the visual appearance of the input image.

**Strengths:**

1. Recasting procedural material generation as a vision-language task, effectively combining visual perception with the generation of structured, editable material programs.
2. Fine-tuning a VLM specifically for procedural material generation, a domain that was previously not widely explored in the vision-language field.
3. Introducing a dataset of 550,000 procedural material samples, which includes not only real-world data but also creatively synthesized samples generated via program-level and parameter-level augmentations. This contribution provides a foundational dataset for further research in this area.
4. Evaluation on both synthetic and real-world images shows that VLMaterial outperforms baselines like BlenderAlchemy and Conditional MatFormer in metrics such as style loss, SWD, and program correctness, demonstrating improvements in visual fidelity and program accuracy.

**Weaknesses:**

1. Limited Novelty in Augmentation Techniques: While the paper presents a large dataset with program-level and parameter-level augmentation, the augmentation techniques themselves rely on GPT-based models and parameter variations, which may not fully capture the variety found in real-world material designs.
2. Despite its strong performance, the model struggles with highly intricate textures, where certain details are either lost or inaccurately represented, as shown in Figure 7. Furthermore, in Table 3, this method underperforms compared to other approaches on out-of-distribution datasets (Substance and real images) when operating under a more constrained sample budget. This limitation may affect its applicability in scenarios that demand high precision.
3. The paper’s evaluation relies heavily on quantitative metrics and in-distribution and out-of-distribution tests. However, given the subjective and artistic nature of material design, the paper would benefit from user studies or feedback from professional material artists. Expert insights could help assess aspects like the usability, editability, and practical value of the generated materials in real production workflows.

**Questions:**

Question 1: In Figure 7, it seems the model has difficulty reproducing certain intricate textures. Are there particular features (e.g., high-frequency details, irregular patterns) that consistently pose challenges? Could you expand on why these features are difficult to capture with your model? It may help to provide an analysis of failure cases or challenging texture types, as well as insights into potential model improvements or data augmentation strategies to better handle intricate textures. This could guide future work and inspire researchers to address these limitations.

Question 2: In Table 3, your model underperforms on out-of-distribution datasets (Substance and real images) with a limited sample budget. Could you elaborate on how the model’s architecture or training process might contribute to this limitation? It might be helpful to explore or discuss possible adjustments, such as adaptive fine-tuning or feature-specific augmentations, to improve generalization on out-of-distribution data under constrained conditions. Additionally, explaining the trade-offs involved in this limitation and suggesting potential remedies would provide a more comprehensive understanding of the model’s practical applicability.

---

> ### Author Response · Authors · 2024-11-21
> **We thank the reviewer for the comprehensive comments. Please find our individual responses below.**
>
> **Limitation of data augmentation (W1)**
>
> Generating procedural materials with novel graph structures and comparable quality to artist-created materials is an open research challenge due to the requirement of validity, diversity, complexity, and rich texture semantics. Our graph structure augmentation method draws inspiration from FunSearch [Romera-Paredes et al. 2024], which leverages the emerging coding capabilities of state-of-the-art LLMs to discover novel solutions to fundamental combinatorics problems. Through a similar evolutionary process, we efficiently explore the combinatorial space of procedural materials and generate diverse programs via asynchronous “genetic crossover” operations. This approach establishes a foundation for future research that could harness the advanced CoT reasoning of next-generation LLMs to produce industry-grade materials.
>
> **Evaluation under a limited budget (W2, Q2)**
>
> The inverse mapping from a material appearance to a node graph constitutes an ill-posed learning problem as node graphs with distinct structures and parameters can represent visually identical material appearances. Therefore, similar to Conditional MatFormer, our VLM may predict multiple valid options in each autoregressive step during inference. This encourages exploration as we randomly sample the next tokens from the predicted distributions, but may require numerous program samples to find the best match. In particular, Hu et al. [2023] generates 150 material graphs using Conditional MatFormer before picking the top-5 graphs for post-optimization.
>
> Fine-tuning the VLM with RL using an image similarity reward will likely reduce the entropy of token predictions and improve generalization in budget-limited scenarios. Nevertheless, RL fine-tuning should carefully balance exploration and exploitation to prevent convergence to local minima. For manuscript revision, we provided additional discussion around Table 3 in Appendix B.

---

### Official Review · Reviewer_dtAr · 2024-10-31

**Soundness:** 3
**Presentation:** 3
**Contribution:** 3
**Rating:** 8
**Confidence:** 4

**Summary:**

This paper tackle the problem of procedural material generation with LLM. The authors first generate Python code conditioned on the input image with LLM, then execute the code in Blender, which can match the input image exactly. This method also facilitates the downstream editing and manual creation.

**Strengths:**

+ This paper proposes a straightforward pipeline for procedural material generation. It trains a domain-specific Vision-Language Model (VLM) through meticulous data collection, processing, and fine-tuning, followed by effective post-processing techniques to address the problem.
+ Both data augmentation and MCMC-based post-processing are validated through qualitative and quantitative results.
+ Using VLM to tackle graphics problems is a promising and intriguing area for exploration, with potential applications across a wide range of domains.
+ The results appear robust, surpassing all baselines, including MatFormer and LLM-based approaches, in both quantitative and qualitative evaluations.

**Weaknesses:**

+ Adapting VLM as a tool for material code generation may not be entirely reasonable, as LLaVa primarily addresses natural images rather than focusing on code generation. It is important for the network design to account for these biases.

+ Additionally, could you provide a detailed user study for artist? Is it possible for this AI tool to substitute certain steps in the artistic creation process?

**Questions:**

+ The paper addresses the challenge of limited material data (~2k) in fine-tuning LLMs through data augmentation. The first question is: what specific techniques were employed in the data augmentation process? Could you summarize these techniques and provide a clearer description?

+ Additionally, the reviewer believes it is quite meaningful to apply LLMs (VLMs) to a broader range of applications, particularly in domains with limited data. Could you provide more evidence regarding the **training/val process** about role of data augmentation?

---

> ### Author Response · Authors · 2024-11-21
> **We thank the reviewer for the thoughtful comments. Please find our individual responses below.**
>
> **Justification for fine-tuning LLaVA (W1)**
>
> While primarily pre-trained on natural languages, VLMs like LLaVA can perform various code generation tasks due to the presence of code blocks in pre-training data. Specifically, LLaVA can generate Blender-style Python programs from single texture images via zero-shot prompting despite some inaccuracies in referencing the Blender API. Thus, we can adapt LLaVA to procedural material generation through supervised fine-tuning. The idea of fine-tuning a VLM for visual program synthesis is also explored by recent work [Zhai et al. 2024] to generate chain-of-thought (CoT) reasoning in task-specific descriptions.
>
> References:
> 1. Simon Zhai et al. Fine-Tuning Large Vision-Language Models as Decision-Making Agents via Reinforcement Learning. Proceedings of NeurIPS 2024.
>
> **Clarification of data augmentation (Q1)**
>
> We have rewritten the relevant paragraphs in Sec. 4.2 to improve clarity. In summary, our data augmentation strategy combines graph structure augmentation and parameter augmentation. The graph structure part employs a similar, evolutionary workflow to FunSearch [Romera-Paredes et al. 2024], which involves performing “genetic crossovers” of randomly selected program pairs using a commercial LLM. The parameter part introduces random parameter perturbations following Hu et al. [2023]. The final augmented dataset contains >550K distinct materials, more than 330x the initial size.
>
> **Role of data augmentation in training and evaluation (Q2)**
>
> The contribution of each augmentation stage to the VLM’s prediction quality is demonstrated in the ablation study (Table 2, Fig. 4, and Fig. 7). Furthermore, we clarify that the augmented programs are not used for qualitative or quantitative evaluation. The Blender test set only contains rendered images of artist-created materials.

---

> ### Comment · Reviewer_dtAr · 2024-11-22
> **Response to the authors**
>
> Thanks for your clarification. I would like to both increase the score and confidence. I wonder if some existed language models specializing in code generation would be better for blender code generation. Can you compare your design with this paper [1]?
>
> Rodriguez et al., StarVector: Generating Scalable Vector Graphics Code from Images, https://arxiv.org/abs/2312.11556.

---

> > ### Author Response · Authors · 2024-11-22
> > **Comparison with StarVector**
> >
> > Thank you for bringing this related work to our attention! We have updated the manuscript to cite the StarVector paper.
> >
> > While both methods use the same VLM architecture design (with a CLIP encoder, an adapter, and a text decoder), the key difference is where the inductive bias is introduced in the model. StarVector uses a code generation model that does not receive multi-modal input. Therefore, their model must acquire image reasoning capabilities from scratch while learning conditional SVG generation. In contrast, our VLM is already pre-trained on VQA data and has partial domain knowledge (such as following the Python syntax and referencing Blender API). The fine-tuning process mostly improves the performance of procedural material generation.
> >
> > In addition, we note that StarVector fine-tunes the entire model, including CLIP, and seemingly without using PEFT adapters for StarCoder. This incurs a higher training cost than our setting. Ultimately, both approaches are viable starting points for VLM fine-tuning with a trade-off between the inductive bias in multi-modal reasoning and code generation.

---

### Official Review · Reviewer_mxJd · 2024-11-04

**Soundness:** 4
**Presentation:** 4
**Contribution:** 3
**Rating:** 8
**Confidence:** 5

**Summary:**

VLMATERIAL generates a procedural node based material pipeline from a given input image. The authors create a dataset (will be open sourced) of pairs of material images and their corresponding graphs by doing clever data augmentation for paramters and node structure. The material graphs are also obtained as Python programs. A VLM is fine tuned on this data to generate a python program to create a material graph from an input image. The method is evaluated on in-distribution (Blender) as well as out-of-distribution data(Adobe substance + real images) and an ablation is performed.

**Strengths:**

* The paper is well-written and motivated. All things are explained clearly.
* The work aims to address a significant issue of directly generating a material graph from an input image as this is highly desriable in CG pipelines. Compared to existing work such as MatFormer and Conditional MatFormer, the results are superior. This is also proven quantitaively in Table 1 against baselines.
* The method creates a large datasset by performing augmentation using an LLM which picks 2 sample programs and then creates a distinct program.
* The VLM in the loop to generate a dataset and also synthesize programs after fine tuning is simple but novel in material generation.
* The ablation results are strong. A test for program correctness is also included.

**Weaknesses:**

* The method is 90% accurate in terms of program correctness compared to other methods which have guaranteed correctness. Have the authors explored methods to reduce LLM hallucination to improve correctness?
* It would good to have a reference for the compution time for different methods

**Questions:**

It would be interesting to see how the complexity of the node setup (number of shader nodes, edges, etc) correlate with the quality of the results and program correctness

---

> ### Author Response · Authors · 2024-11-21
> **We thank the reviewer for the encouraging comments. Please find our individual responses below.**
>
> **Additional results (W2, Q1)**
>
> We included the inference time of each method (Table 4) and added a scatter plot (Fig. 8) in Appendix B to illustrate the impact of program complexity on prediction quality and program correctness. Most materials with moderate complexity show comparable style losses and correctness. Interestingly, both metrics are worse for simple materials with fewer than 4 nodes, which is largely due to non-existent or suboptimal choices of custom node groups. Program correctness also declines for materials with more than 22 nodes or 2,000 tokens, implying a higher chance of errors in more intricate materials. As mentioned in the general response, implementing inference-time syntax checking to mask out invalid tokens can guarantee the correctness of generated programs.

---

> ### Comment · Reviewer_mxJd · 2024-11-23
>
> Thanks for the clarifications, I'm happy with the response.
>
> I would like to further point out that there is a very recent work on the same exact problem of procedural material generation-- https://dl.acm.org/doi/10.1145/3687979.
> While I don't expect this work to be compared or the authors to know about this given the work released after ICLR submission. However, it might be nice to have this in the related work anyway given the relevance.

---

> > ### Author Response · Authors · 2024-11-24
> > **Reference to a recent publication**
> >
> > Thank you for suggesting this recent publication! We have included the paper in the related works section and also referenced it in future works considering its relevance to RL fine-tuning.

---

### Official Review · Reviewer_fAs1 · 2024-11-07

**Soundness:** 4
**Presentation:** 3
**Contribution:** 3
**Rating:** 8
**Confidence:** 4

**Summary:**

This paper proposes a framework for finetuning a VLM for generating procedural materials that matches an input image. A new procedural material dataset collected and curated from various sources, and then augmented with an LLM based approach. Output generated from the finetuned VLM also undergoes a MCMC-based gradient free optimization step to bring it closer to the input image. The results are impressive both qualitative and quantitatively, and with better generalzation capacities.

**Strengths:**

- Personally, I always enjoy seeing papers that show "foundation models work on X". Even if there is minimal contribution architecture and algorithm wise, it still provide me with insight about what kind of tasks can benefit from large models. This paper adds even more evidence about the viability of applying LLM/VLMs to procedural generation (and graphics in general), and I find this message important.
- Beyond the rather straightforward application of LLM + finetuning, considerable works as also being done here for creating the finetuning data. There are valuable insights on how data is cleaned and augmented, and the LLM based approach for creating new samples from two examples seem generally applicable for anything that involve visual programs. The dataset itself can also be immensely useful, especially when the alternative is not publicly/easily accessible.
- Solid insights on how to work with procedural material graphs that a not differentiable. The MCMC based approach is reasonable, appears to be working from the ablations and examples. It is also nice (and a bit nolstalgic) to add some exposure to a classic set of optimization methods and show that they are very viable in certain cases.
- The results look good. I do have a good amount of issues with some results but overall, it does appear to be that there are cases where the proposed method works clearly better than alternatives. Ablation is also solid and effective.

**Weaknesses:**

- Also I do appreciate the message that "VLM works for procedural materials", arguably the novelty is limited. This is one of the lower hanging fruits, and a good amount of work is engineering-centric, making the impact of this work potentially limited. It probably also makes this paper less suitable for a venue like ICLR, since there isn't too much contribution learning-wise. I would definitely love to see this at a graphics venue a bit more.
- Following the previous point - even for paper that's mostly "apply VLM to X", the amount of insights in the paper is still on the relatively low end. Most of the discussion is around the data. They are important, and I do consider the data portion a strength. However, I feel that there's missed opporunities here in further invesgitating how to best apply VLMs to this problem, especially since the supervision is entirely token level, without visual feedback (comparing the generated material to the input). E.g. there are many known limitations with VLM/LLMs that make them not perfectly suitable for directly outputing complex visual programs with lots of parameters that are non-trivial to interpret on a purely language level. How to design something that alleviate such issues? Does fine tuning take care of most of it or do we need to more carefully design/modify the language and the prompts? What part of the output do the model struggle the most with? Is there a more intuitive explanation of why failure cases like those in Figure 7 happen, and what part of the output contribute the most to the discrepancy between the output material and the image (e.g. it does appear that many BSDFs are quite off?)? Discussions along these lines would be very helpful both for people who want to use this approach, or for future researchers that might build upon this.
- My standard on quality of results is definitely higher on this one, due to the rather limited amount of technical insights. And while the results look good overall, they are still quite far from matching the input image, even among the few qualitative examples provided and after the post-processing step.

**Questions:**

I am overall positive on this one. Despite the rather limited novelty and lack of deeper exploration, there is still solid data contribution, a good amount of insights, and solid results. I won't champion this paper in its current state, but more "insights" (see weakness section above) will most likely move me towards a more positive stance on this paper.

---

> ### Author Response · Authors · 2024-11-21
> **We thank the reviewer for the insightful comments. Please find our individual responses below.**
>
> **Sources of discrepancies between the output material and the image (W2)**
>
> For most challenging, out-of-distribution test images, the limited expressiveness of generated graph structures is a primary source of visual discrepancies. Other minor mismatches in color and spatial transformation are typically attributed to parameter predictions. Additionally, in these challenging scenarios, the VGG-based style loss may not perfectly correlate with human perception, which affects the choice of the best predicted sample. As our work focuses on prediction quality, enhancing the expressiveness of the Blender node graph system using more custom node groups from artist-created materials and incorporating visual feedback through RL fine-tuning are both promising directions for future improvements.

---

### Author Response · Authors · 2024-11-21
**General response**

We thank all reviewers for taking the time and effort to review our submission and leave constructive comments. We appreciate their acknowledgment of our contribution, cross-domain impact, and solid improvement from baselines.

Please find our responses to some shared questions and concerns below. Answers to reviewer-specific comments are posted in corresponding threads. We have also carefully advised the manuscript to make further clarifications and include additional results as requested by the reviewers. We hope that we have adequately addressed all the raised issues and we are happy to provide more information as needed.

**User study on generated materials (Reviewer dtAr, bStb)**

We conducted a user study on the usability of algorithm-generated procedural materials among professional artists and researchers experienced in the topic. The user study involves comparing materials generated from our method and BlenderAlchemy in visual quality and node graph editability without knowing the source algorithm. The participants are also asked to indicate their preference and rate the usability of each generated material on a scale from 1 to 10 based on how much it substitutes an artist’s creation process. We collected 16 responses in total (from 8 artists and 8 researchers). As such, we hope the reviewers understand the difficulty in finding available, qualified participants within a short period.

The design and results of our user study are provided in Appendix C. In more than 90% of the test cases, the participants prefer our generated materials to BlenderAlchemy in visual and node graph quality. Our method achieves an average usability rating of 6.8, indicating that participants generally consider VLM-generated materials efficient starting points for post-editing. In addition, some artists praised our method for the interesting results and appreciated the better reusability of simpler graph structures in production environments.

**Better application of VLM fine-tuning (Reviewer fAs1, mxJd)**

The challenge of generating complex visual programs using a VLM can be approached from three orthogonal directions based on our supervised fine-tuning methodology:
- **Additional training data**: Our ablation study (Table 2) shows that data augmentation in the graph structure space and the parameter space substantially improves prediction quality and program correctness. Therefore, it will be beneficial to add more high-quality artist-created materials to the dataset and augment the training data with more diverse, complex programs using state-of-the-art commercial LLMs. The augmentation prompt can also be updated to better harness the chain-of-thought (CoT) reasoning of next-generation LLMs like OpenAI o1.
- **Domain-specific language**: It is possible to represent Blender procedural materials with a more compact domain-specific language (DSL) based on handcrafted rules or library learning. This will shorten the transpiled programs considerably compared with the Python language and simplify VLM fine-tuning. Moreover, an inference-time syntax checker can be implemented for the DSL to mask invalid tokens and guarantee the correctness of generated programs.
- **Visual feedback**: Incorporating visual feedback through RL fine-tuning is an effective approach to enhance visual quality as employing an image-space reward function bridges the gap between token-level accuracy and visual similarity. Incorporating a DSL for procedural materials further reduces token sequence lengths and allows the reward to propagate in fewer steps.

Overall, the best application of VLM fine-tuning to our task should combine all three techniques organically. To provide more insights into better VLM fine-tuning strategies, we have added a paragraph for future works and expanded on other relevant discussions in the revised paper.

**Explanation for failure cases (Reviewer fAs1, bStb)**

The representation capability of a procedural material is constrained by Blender’s built-in node types and existing artist-created custom node groups. Fig. 7 shows a few examples where the intricate geometric patterns in an out-of-distribution input image exceed the expressiveness of Blender materials within a reasonable node graph size (e.g., we use a limit of 30 nodes). In such cases, our model may struggle to predict the best material graph structure to approximate the input texture patterns. Extending the node graph system with custom node groups from a broader range of artist-created materials will alleviate the limitation in expressiveness. An alternative is to increase the graph size limit, although this requires additional engineering efforts in data augmentation to generate more feature-rich materials using the LLM. Introducing a DSL could be helpful in this scenario as it allows the VLM to generate more sophisticated node graphs with shorter programs. We have revised the limitations section accordingly to clarify the reasoning behind failure cases.

---

### Meta-Review · Area_Chair_9xmU · 2024-12-23

**Metareview:**

This paper proposed an approach for generating procedural materials, which are represented as functional node graphs for photorealistic material design. The key idea is to approach the problem as a Python code generation task using a large-pretrained vision-language model (VLM). In addition, the authors provided an open-source procedural material dataset. AC confirms that all reviewers appreciate the problem configuration and that the proposed approach is effective. After the discussion phase, the constructive comments are properly applied to the revision. AC also agrees the merit of the proposed approach and the newly designed dataset.

**Additional Comments On Reviewer Discussion:**

This paper received a strong score after the discussion phase, and it reached a strong consensus for the paper's acceptance. In general, reviewers ask questions about the user study on the generated materials, another approach for better VLM fine-tuning, and explanations for failure cases. The authors provided proper feedback on these common questions and made a revision in the submitted paper.

Regarding the other questions, reviewer fAs1 mentioned discrepancies between the output material and the image. The reviewer states that the revision is acceptable and suggests RL-based visual feedback and analysis of the program size as future work. The reviewer mxjd gave strong scores and asks about additional results. The reviewer also recommends another paper using an RL-based fine-tuning approach. The authors provided new material in the appendix and included the referred paper in the related work. The reviwer dtAr gave the short comment that are properly addressed in the common questions. The reviewer dtAr explicitly mentions comparing the proposed approach with an ArXiv paper, which is not reasonable during the rebuttal phase. The reviwer bStb mentions about limitation of data augmentation and limited amount of evaluation. The reviewer raised the score after reading the authors' feedback.

In general, AC confirms that the discussion was constructive, and the new materials are properly applied to the revised paper.

---

### Decision · Program_Chairs · 2025-01-22

Accept (Spotlight)